# Secrets of Flavonoid Synthesis in Mushroom Cells

**DOI:** 10.3390/cells11193052

**Published:** 2022-09-29

**Authors:** Jan Pukalski, Dariusz Latowski

**Affiliations:** Jagiellonian University, Faculty of Biochemistry, Biophysics and Biotechnology, Department of Plant Physiology and Biochemistry, Gronostajowa 7, 30-387 Kraków, Poland

**Keywords:** mushrooms, fruiting bodies, mycelia, flavonoids, biosynthesis, HPLC, gene expression, metabolites

## Abstract

Flavonoids are chemical compounds that occur widely across the plant kingdom. They are considered valuable food additives with pro-health properties, and their sources have also been identified in other kingdoms. Especially interesting is the ability of edible mushrooms to synthesize flavonoids. Mushrooms are usually defined as a group of fungal species capable of producing macroscopic fruiting bodies, and there are many articles considering the content of flavonoids in this group of fungi. Whereas the synthesis of flavonoids was revealed in mycelial cells, the ability of mushroom fruiting bodies to produce flavonoids does not seem to be clearly resolved. This article, as an overview of the latest key scientific findings on flavonoids in mushrooms, outlines and organizes the current state of knowledge on the ability of mushroom fruiting bodies to synthesize this important group of compounds for vital processes. Putting the puzzle of the current state of knowledge on flavonoid biosynthesis in mushroom cells together, we propose a universal scheme of studies to unambiguously decide whether the fruiting bodies of individual mushrooms are capable of synthesizing flavonoids.

## 1. Introduction

Fungal cells are known to produce a variety of biological compounds. They are a source of medicinal substances, such as antibiotics [1,2,3], anticancer drugs [4,5] and even psychoactive compounds such as psilocybin, the application of which is controversial but postulated by some scientists to be useful in mental disorder treatments [6]. Fungal cells are also capable of synthesizing large polymers such as melanins [7,8] and chitosan [9], which are considered promising biomaterials [10,11,12], and notably, whole-mycelium-based composites are intensively studied as, among others, eco-friendly building materials [11,13]. A diverse group of fungal metabolites is the sesquiterpenes, which may be characterized by antifungal, antibacterial, cytotoxic, anti-inflammatory and anticancer properties [14]. As sesquiterpenes can be non-volatile or volatile [15], fungal volatile organic compounds (fVOCs) constitute another large group of ecologically important compounds, which is well described in Inamdar et al.’s [16] and Guo et al.’s [17] articles. Fungal cells are also a rich source of pigments, including not only the previously mentioned melanins but also oxopolyenes, quinones, anthraquinones, naphthoquinones, hydroxyanthraquinones and azaphilones [18]. Some pigments may also be valuable food ingredients, for example, carotenoids and flavonoids [19,20]. Fungi are considered a promising source of nutrients. Barzee et al. [21] stated that even microscopic filamentous fungi such as *Fusarium venenatum*, *Aspergillus oryzae* and *Rhizopus oligosporus* mycelia, after their proper preparation, may be consumed and become foods of the future. Interestingly, the mentioned fungi are characterized by high protein contents, with 48.0, 45.7 and 49.7 g per 100 g of dried weight, respectively. They also contain all essential amino acids required by humans and are a source of dietary fiber, unsaturated fatty acids, carbohydrates, macroelements such as calcium, potassium, and phosphorus and microelements involving, i.a., zinc, of which an especially rich source is *F. venenatum*, with 30.4 mg/100 g of dried weight [21]. However, today, as has been the case for centuries, more attractive to consumers are edible mushrooms such as *Agaricus bisporus*, *Pleurotus sajor-caju* or *Pleurotus giganteus*, the fruiting bodies of which are characterized by relatively high protein–fat ratios of 14.1:2.2, 37.4:1.0 and 17.7:4.3, respectively, and are a familiar, accepted form [22]. Moreover, nowadays, mushroom fruiting bodies are intensively studied as food sources, and detailed information about edible species, nutrient compositions, the digestibility of particular components and taste attributes can be found elsewhere [23,24,25].

However, what is meant by the term ‘mushroom’? There are two definitions: the first one describes mushrooms as macroscopic fruiting bodies only, and the second one (in our opinion, the better one) is a species that produces macroscopic fruiting bodies and mycelia that grow through the nutrient substrate. According to *Encyclopedia Britannica*, mushrooms are defined as ‘the conspicuous umbrella-shaped fruiting body (sporophore) of certain fungi, typically of the order Agaricales in the phylum Basidiomycota but also of some other groups. Popularly, the term mushroom is used to identify the edible sporophores; the term toadstool is often reserved for inedible or poisonous sporophores. There is, however, no scientific distinction between the two names, and either can be properly applied to any fleshy fungus fruiting structure’ (https://www.britannica.com/science/mushroom, accessed on 27 May 2022). Another description from *Merriam-Webster Dictionary* is ‘an enlarged complex aboveground fleshy fruiting body of a fungus (such as a basidiomycete) that consists typically of a stem bearing a pileus. Especially: one that is edible’ (https://www.merriam-webster.com/dictionary/mushroom, accessed on 27 May 2022). In addition, some typical scientific sources define mushrooms as fruiting bodies only; for example, ‘Mushrooms are macroscopic fruiting bodies produced by ascomycete and basidiomycete fungi during their sexual reproduction cycles’ [26]. However, mushrooms are also commonly considered a group of fungal species called macrofungi, and according to this, two other descriptions should be presented. The first one, taken from the *Cambridge Dictionary*, describes a mushroom as ‘a fungus with a round top and short stem. Some types of mushroom can be eaten’ (https://dictionary.cambridge.org/pl/dictionary/english/mushroom, accessed on 27 May 2022). The second one, proposed by Chang and Miles in 1992 [27], states that a mushroom is ‘a macrofungus with a distinctive fruiting body, which can be either hypogeous or epigeous, large enough to be seen with the naked eye and to be picked by hand’. The two definitions, especially the last one, seem to be consistent with the distinctions made in articles by Carvajal et al. [28] and Fijałkowska et al. [29], where a mushroom is considered a fungal species that can produce both mycelium, which grows through the soil, touchwood or any other substrate, and fruiting bodies. According to this, macrofungi of basidiomycetes, such as *Auricularia* spp. or polypores, and ascomycetes, such as *Morchella* spp., *Gyromitra* spp. and *Cordyceps* spp., as well as hypogeous tubers, should also be termed mushrooms.

As was mentioned earlier, pigments, such as carotenoids and some flavonoids (since not every flavonoid is a pigment), in fungal cells are desirable as food ingredients. Some microscopic species from the genera *Xanthophyllomyces*, *Rhodotorula*, *Sporidiobolus*, and *Phaffia*, as well as the edible mushroom *Cantharellus cibarius* (Fries), are able to synthesize carotenoids [30,31]. Flavonoids can be synthesized by microscopic fungi, which was proven for *Aspergillus* sp. YXf3 using NMR method [32], but the ability of mushroom fruiting bodies to produce flavonoids seems to still be shrouded in a veil of secrecy. In 2016, Gil-Ramírez et al. [33] after versatile analyses stated that mushroom cells were not capable of producing flavonoids. In 2020, Mohanta [34], in direct response to Gil-Ramírez et al. [33], challenged the statement and showed a list of fungi whose cells contain genes connected to flavonoid synthesis. However, this extensive list mainly included microscopic fungi, and our analysis showed there were no mushrooms with the full set of genes that were postulated as essential in the flavonoid biosynthesis pathway. Thus, the question of whether mushroom cells are capable of synthesizing flavonoids needs to be explained.

## 2. Are Mushrooms Capable of Producing Flavonoids?

Flavonoids are mainly considered plant secondary metabolites [35,36] that play protective roles against UV radiation [37] and reactive oxygen species (ROS) [38]. They are essential in plant–pollinator interactions [39] and protect the plant cell host from pathogenic microbes, among which plant–fungi interactions and the antifungal properties of flavonoids are especially well described [40,41,42,43,44,45,46,47,48,49]. Thus, flavonoids, despite the positive role that they play in plant interactions with mycorrhizal fungi [50,51], are generally described as harmful to fungal cells. However, scientists are still focused on searching for fungi that produce flavonoids, especially mushrooms that evolve edible fruiting bodies [31,52,53], due to the health-promoting properties of these compounds [54,55,56,57,58,59,60,61,62,63]. Additionally, from the scientific point of view, the role of flavonoids in fruiting bodies is exceedingly interesting due to the fact that, except for hypogeous species, fruiting body cells are exposed to biotic and abiotic stresses that differ from those of mycelia located in the soil or touchwood.

In 2016, a group of scientists led by Alicia Gil-Ramírez published the article ‘Mushrooms do not contain flavonoids’ [33]. The authors decided to conduct three types of analyses: first, they studied 136 articles concerning the detection of flavonoids in mushrooms; secondly, they analyzed databases to find protein sequences of enzymes essential for flavonoid synthesis; and thirdly, they studied the mushrooms themselves for flavonoid presence. After a literature review, the researchers realized that in almost 91% of publications concerning flavonoid detection in mushroom cells, colorimetric methods dedicated to plant cells were used. Those analyses were based on the procedure proposed by Christ and Müller in 1960 [64], but in 2014, it was challenged by Pękal and Pyrzynska [65], who showed that, even in plants, the method is not specific enough to serve as a total flavonoid content measurement procedure. Gil-Ramírez et al. [33] also mentioned 13 articles that revealed flavonoid presence in the mushroom fruiting bodies of, among others, *Pleurotus ostreatus* and *Agaricus bisporus* when methods more advanced than colorimetric methods, such as HPLC, GC, UPLC with DAD, UV or MS detection, were used. However, also in this case, they stated that flavonoids could not be synthesized by fungal cells but were absorbed by mushrooms from neighboring or mycorrhizal plants [33].

To further investigate the ability of mushroom cells to produce flavonoids, Gil-Ramírez et al. [33] conducted the second stage of their research and used Pfam and Blast search tools to find gene sequences homologous to the gene sequences of the three enzymes considered crucial for flavonoid synthesis in plant cells. These enzymes are: phenylalanine ammonia-lyase (Pal), which converts L-phenylalanine to trans-cinnamic acid and ammonia (Figure 1A); chalcone synthase (Chs), which catalyzes the condensation of coumaroyl-CoA and three units of malonyl-CoA to naringenin chalcone (Figure 1B); and chalcone isomerase (Chi), which mediates the cyclization of naringenin chalcone to the flavanone (Figure 1C) [66].

Among all fungal species with genomes recorded in databases as of 2015, the researchers found genetic sequences encoding Pal-like enzymes in four mushroom species (*Agaricus bisporus*, *Tricholoma matsutake*, *Amanita muscaria* and *Pleurotus eryngii*), seven microscopic fungal species (*Pseudozyma brasiliensis*, *Rhodosporidium toruloides*, *Rhodotorula mucilaginosa*, *Rhodotorula glutinis*, *Sporidiobolus salmonicolor*, *Penicillum roqueforti* and *Fusarium graminearum*) and one lichen species (*Letharia vulpine*). Subsequently, they searched for sequences homologous to the sequence encoding Chs in *Vitis vinifera*, but no mushroom species with significant homology (the highest homology of 17% was calculated for the *Pleurotus ostreatus* C15 strain) were found. Similarly, sequences homologous to the gene encoding Chi were not found among mushrooms either [33].

In the third stage of their work, the authors studied flavonoid presence in the fruiting bodies of: *Pleurotus eryngii*, *Pleurotus ostreatus*, *Lentinus edodes*, *Pleurotus pulmonarius*, *Amanita ponderosa*, *Agaricus bisporus*, *Agaricus blazei*, *Lactarius deliciosus*, *Lyophyllum shimeji*, *Craterellus cornucopioides*, *Auricularia judea*, *Amanita caesarea*, *Hydnum repandum*, *Pholiota nameko*, *Hericium erinaceus*, *Morchella conica*, *Ganoderma lucidum*, *Lepiota procera*, *Calocybe gambosa*, *Boletus edulis*, *Cantharellus cibarius*, *Agrocybe aegerita* and *Grifola frondosa*. Additionally, they cultivated the commercially available *Pleurotus ostreatus* K8 strain on media supplemented with onion waste to study flavonoids in fruiting bodies and on media containing flavonoids and phenolic compounds, such as quercetin, luteolin, caffeic acid, luteolin-7-O-glucoside or hesperidin, to analyze flavonoids in mycelia. The mentioned cultures were used to verify whether fungi, both fruiting bodies and mycelia, were able to absorb flavonoids from the enriched media. The authors decided to compare the results of colorimetric methods used in other works with the results obtained from HPLC analyses. For this purpose, they applied the procedure described by Ramírez-Anguiano et al. in 2007 [67] for the calculation of ‘total phenolics content’ and the method used by Choi et al. in 2006 [68] to determine the ‘total flavonoids’ quantity, and then they compared the results with data obtained from analyses conducted on HPLC-DAD and HPLC-DAD-ESI/MS systems in two different labs. The authors revealed that colorimetric methods provided positive results suggesting that flavonoids were present, but HPLC analyses showed that no flavonoids were detected in the fruiting bodies or mycelia of the studied fungi. The researchers [33] decided to verify which components in fungal hyphae could be the reason for the false positive results of colorimetric analyses. They revealed that the UV-Vis spectra of two biocomponents abundant in fungi—ergosterol, the main fungal sterol, and L-dihydroxyphenylalanine—treated in accordance with ‘total flavonoids’ protocols were characterized by absorption bands at wavelengths of 415 and 510 nm, which are analytical bands in ‘total flavonoids’ measurement methods. Thus, the mentioned compounds could be falsely detected as flavonoids.

The researchers also provided interesting information concerning the growth and development of fungi exposed to flavonoids. One of the outcomes was that luteolin and luteolin-7-O-glucoside reduced the growth of *P. ostreatus* K8 mycelium. The reduction level was correlated with the concentrations of the mentioned compounds in the medium, and the aglycone form caused a stronger inhibition of mycelial growth than the glucoside form (unfortunately, the last statement was not supported by graphical or statistical analysis) [33].

Gil-Ramírez et al.’s [33] article is a source of valuable information about mushrooms and flavonoids. However, the opinion that mushroom cells in general cannot produce but may absorb and accumulate flavonoids was derived from the analysis of databases containing just a few fully sequenced mushrooms and many that were only partially sequenced. Additionally, the authors analyzed the fruiting bodies of 24 species, among which only *P. ostreatus* K8 was named by strain. Thus, the opinion based on those results should be strictly limited to the studied mushroom group only, and with such scarce data, it should not be extended to the rest of the species and strains, distancing researchers from knowing the actual ability of mushroom cells to produce flavonoids.

In 2020, Tapan Kumar Mohanta [34] published the article ‘Fungi contain genes associated with flavonoid biosynthesis pathway’, which was a direct response to Gil-Ramírez et al.’s [33] paper. The mentioned authors [34] conducted impressive, thorough research and found many fungal species and strains whose cells contain gene sequences encoding not only Pal, Chs and Chi but also the genes of many other enzymes specifically involved in flavonoid synthesis. Mohanta found sequences homologous to, among others, the genes of dihydroflavonal-4-reductase, naringenin 3-dioxygenase, rutin-alpha-L-rhamnosidase and isoflavone reductase, and finally stated that ‘it is not correct to say fungi/mushroom do not contain enzyme associated with biosynthesis of flavonoids’ [34]. However, after screening records from ‘Appendix A. Supplementary material’ prepared by Mohanta [34], we realized that only two mushroom species, *Armillaria gallica* and *Armillaria solidipes*, contained nucleotide sequences homologous to those encoding Pal isoforms. However, they did not contain sequences homologous to sequences encoding Chs or Chi. In addition to the two species mentioned above, among the fungi listed in Appendix A, there were no mushrooms defined that contained the genes of all three enzymes postulated as crucial for flavonoid synthesis. There were not even any mushrooms that contained both genes encoding Chs and Chi. *Lentinula edodes* and two edible hypogeous mushrooms, *Tuber borchii* and *Tuber magnatum*, contained genes homologous to sequences encoding Chi, but none of the mushrooms possessed sequences encoding Chs. Also found were gene sequences of putative quercetin 2,3-dioxygenase in *Termitomyces* sp. J132 and caffeoyl-CoA O-methyltransferase in *Auricularia subglabra* and two *Tolypocladium* species. The article and supplementary material provided by T. K. Mohanta [34] are valuable sources of information about fungal genome sequences related to enzymes involved in flavonoid synthesis; however, it cannot be considered a refutation of Gil-Ramírez et al.’s [33] statements. Moreover, it cannot be accepted as proof that mushroom cells possess essential genes for flavonoid synthesis either, because Mohanta’s paper [34] described no mushroom species that present gene sequences of all three of the enzymes Pal, Chs and Chi, or at least Pal and Chi. Surprisingly, in the supplementary material [34], there were also no microscopic fungi presented that possessed all three required sequences. After reviewing the two mentioned articles, the question of whether mushrooms are capable of producing flavonoids remains unanswered.

## 3. Current Achievements and Future Prospects

The articles by Gil-Ramírez et al. [33] and Mohanta [34] started an interesting discussion about the abilities of mushroom cells to produce flavonoids and encouraged other scientists to conduct more detailed experiments in this field. The first conclusion that can be drawn from the mentioned articles is that colorimetric methods should be replaced by more advanced methods, such as those utilizing HPLC and HPLC-MS systems [33]. Gil-Ramírez et al. [33] also addressed an important issue, which is the possible absorption of flavonoids from the local environment or plants neighboring mushrooms. It was proven that fungi are able to absorb not only simple saccharides, amino acids and minerals but also molecules such as vitamins, i.e., thiamine, thiamine phosphates, pyridoxine or pantothenic acid, and even siderophores using specific transporters [69,70]. Slana et al. [71] applied LC-MS systems to analyze whether fungal mycelium is able to absorb flavonoids from the environment. The researchers revealed that Rhizopus nigricans is able to absorb compounds such as biochanin A and transform them into less toxic glycosides. Gonzales et al. [72] studied the *Rhizopus*
*azygosporus* Yuan et Jong (ATCC 48108) strain exposed to quercetin and naringenin aglycones. Analyses conducted using LC-MS and UPLC-MS systems revealed that both quercetin and naringenin were absorbed by fungal mycelia, but the first one was partially transformed into quercetin glucoside, diglucoside, glucosyl-sulfate and sulfate, and the second one was fully transformed into eriodictyol glucoside and eriodictyol sulfate. Interestingly, the growth of the fungus was not inhibited, and some of the metabolites accumulated intracellularly [72]. Thus, it can be concluded that the absorption of flavonoids in the fungal kingdom occurs and is experimentally fully proven. To eliminate the possibility of absorption, when studying the capability of mushroom cells to synthesize flavonoids, the best solution is to cultivate mushrooms on artificial media or spawn [73,74] with a well-controlled composition. According to this, Hasnat et al. [75] and Lin et al. [76] published articles about flavonoids and other phenolics in *Ganoderma lucidum* and *Pleurotus eryngii*, respectively. They used HPLC systems and detected flavonoids in the fruiting bodies of the mentioned mushrooms. However, cultivated media were enriched with rice or rice bran, which are known to be sources of flavonoids [77,78]. The researchers [75,76] did not state that the detected compounds were synthesized by the mushroom cells, but Gil-Ramírez et al. [33] listed both publications [75,76] in a section that presented articles revealing flavonoid presence in mushrooms. Gil-Ramírez et al. [33] did not state that the mentioned mushrooms produced flavonoids, but this could be easily misinterpreted by the readers. However, the content of flavonoids in the fruiting bodies of the studied species [75,76], even if they are not synthesized by mushrooms, fits well with the theory of the possible absorption of flavonoids by fungi. This issue is important, and so far, only Gil-Ramírez et al. [33] have studied the absorption and accumulation of flavonoids in the mycelia and fruiting bodies of mushrooms. Thus, in experiments testing fungal cells’ abilities to produce flavonoids, it is recommended to use cultivation media that do not contain the mentioned compounds. On the other hand, not every mushroom can be cultivated under laboratory conditions; however, although obtaining fruiting bodies in vitro can sometimes be impossible, the cultivation of mycelia seems to be easier and more successful. Meng et al. [79] recently published interesting results from transcriptomic and metabolomic analyses of flavonoid-biosynthesis-related pathways in cells of the edible mushroom *Auricularia cornea* cultivated on media with different substrates. They analyzed the expression of genes engaged in flavonoid biosynthesis and applied non-targeted metabolomic analysis using an LC-MS system to find metabolites of the flavonoid biosynthesis pathway. The researchers proved, among others, that metabolites connected to coumarin, phenylpropanoid and isoflavonoid biosynthesis pathways were produced in *A. cornea*, and especially interesting was the detection of the isoflavones biochanin A, formononetin and coumestrol. They also revealed that the addition of the leachate of Korshinsk peashrub (*Caragana korshinskii*) to the complete yeast medium (CYM) could increase the flavonoid content by up to 23.6% in comparison with the amount obtained after the cultivation of *A. cornea* on the control CYM medium [79]. Recently, studies concerning flavonoid presence and synthesis in *Sanghuangporus baumii* cells were published by Wang et al. [80]. The researchers applied UPLC–MS/MS analysis and identified eighty-one flavonoid compounds from nine groups: flavanols, flavanonols, flavones, flavonols, flavanones, chalcones, isoflavones, isoflavanones and proanthocyanidins in numbers of 10, 4, 25, 23, 5, 2, 8, 1, and 3, respectively. The flavanonol pinobanksin and the glycosylated and acylated forms of the isoflavone genistein were detected, and the flavonol kaempferol, linked to the pro-health properties of *S. baumii*, was identified as well. Wang et al. [80] also revealed that an increased aeration time promoted the synthesis of flavonoids, which would be especially valuable for the industrial production of the compounds. Due to the variety of flavonoids detected in mushroom mycelia [79,80], where the risk of the absorption of the compounds does not appear, the biosynthetic pathways of the compounds are especially interesting. In the genome of *A. cornea*, Meng et al. found four homologous genes of *pal*, three homologs of the polyphenol oxidase-encoding gene (*ppo*) and two genes homologous to *chi* [79]. Wang et al. [80], in their studies of *S. baumii*, detected eighty-one flavonoids, but only four genetic sequences homologous to sequences encoding plant enzymes involved in flavonoid biosynthesis (*pal*, *chi*, *4-coumarate-CoA ligase* (*4cl*) and *isoflavone reductase* (*ifr*)) were found. The lack of, for example, sequences of *cinnamate 4-hydroxylase* (*c4h*), *flavanone-3-hydroxylase* (*f3h*) or *flavonol synthase* (*fls*), as well as *chs*, led the researchers to the conclusion that, in the case of mushrooms, in flavonoid biosynthesis, other enzymes of the same superfamilies as the absent ones are involved. The authors also revealed that the overexpression of *sbPal* increased the amounts of flavonoids accumulated in mycelia [80]. The researchers [79,80] conducted metabolomic and gene expression analyses, which proved that the synthesis of flavonoids occurred. The articles by Meng et al. [79] and Wang et al. [80] provided a great deal of interesting information about fungal biochemistry and proved the biosynthesis of flavonoids in the cells of *A. cornea* and *S. baumii* mushroom species. Unfortunately, in both papers, the mycelia, not the fruiting bodies, were studied.

According to Mohanta [34], ‘Mushroom is a fruiting body of the fungi and hence it contains all the genomic architecture of its genome. The genomic architecture of the fruiting body is not separated from its mycelia’. Mohanta interprets the term mushroom as the fruiting body only, not as a species that produces mycelium and a macroscopic fruiting body. However, this distinction can be neglected here, and it is understood that he stated that the mycelia and fruiting bodies possess the same set of genes. This statement is correct, but the question about differences in gene expression in mycelia and fruiting bodies appears. Lin et al., in 2017 [81], published an article in which the differential expression of genes encoding terpenoids in both structure types was proven and led to the synthesis of terpenes such as terpineol, thymol, limonene or α-cadinol exclusively in the fruiting bodies of the *Antrodia cinnamomea* mushroom. Song et al., in 2018 [82], listed a variety of differentially expressed genes (DEGs) in *Lentinula edodes* mycelia and fruiting bodies. Wang et al. [83] and Fu et al. [84] published articles with analyses of DEGs in different stages of the development of mushroom cells in fruiting bodies. According to this, it is recommended to analyze gene expression and metabolite presence in fruiting body cells, not in mycelia. Additionally, we suggest analyzing different developmental stages of fruiting bodies to indicate the particular stage in which flavonoid synthesis and accumulation occurs or is the most efficient.

During the literature review, two articles concerning flavonoid presence in *Inonotus sanghuang* (recently renamed *Sanghuangporus sanghuang*), an Asian medicinal mushroom, especially drew our attention. In the first one, researchers led by Liu used an LC-MS-IT-TOF system and revealed the presence of isorhamnetin, quercitrin, rutin and quercetin in fruiting bodies obtained as environmental samples [85]. In the second article by Shao et al. [86], the expression of genes associated with secondary metabolism in the *Sanghuangporus sanghuang KangNeng* strain was studied. The authors studied four different developmental stages of the mushroom: 10- and 20-day-old mycelia, as well as 1- and 3-year-old fruiting bodies. The researchers analyzed genes involved in the biosynthesis of polysaccharides, triterpenoids and flavonoids. Interestingly, they suggested that mycelial stages, based on gene expression analysis, are more effective flavonoid producers than fruiting bodies. The researchers analyzed genes related to flavonoid biosynthesis and found sequences of *chalcone isomerase 1* (*chi1*), *flavonoid-3′*, *5-hydroxylase* (*f3′5′h*), *f3′H -flavonoid-3′-hydroxylase* (*f3′h*), *flavanone-3-hydroxylase 1* (*f3h1*), *flavanone-3-hydroxylase 2* (*f3h2*) and *flavonol synthase* (*fls*). What is worth mentioning here is that *f3h* homologs and *fls* were not found in the genome of *S. baumii* [80]. It was proven that the genes were activated in the fruiting body cells of the *S. sanghuang KangNeng* strain cultivated in culture media free of flavonoids under controlled laboratory conditions. Interestingly, the researchers realized that genes encoding Chs were not present in the cells of this mushroom species [86], which is consistent with Mohanta’s [34], Meng et al.’s [78] and Wang et al.’s [80] studies. Results presented by Shao et al. [85] suggested the presence of an alternative to the Chs enzyme in the biosynthesis pathway of flavonoids in mushroom cells in comparison to plant cells. Wang et al. [80] suggested that the Chs function may be served by polyketide synthase (Pks), and they announced further studies to determine it.

Despite the incomplete understanding of the biosynthetic pathways of mushroom flavonoids, Meng et al. [79] and Wang et al. [80] proved the ability of the mycelial cells of mushrooms to produce flavonoids. The combined results presented by Liu et al. [85] and Shao et al. [86] showed *S. sanghuang* to be, according to our best knowledge, the first mushroom species in which fruiting bodies were studied by means of metabolomics and gene expression, and the analyses yielded positive results in the case of flavonoid synthesis. If Liu et al. [85] had conducted additional analyses and proved that genes conditioning flavonoid biosynthesis pathways were actively expressed, or if Shao et al. [86] had applied instrumental analysis methods, such as those using HPLC systems, and revealed the presence of the final products (flavonoids), the *S. sanghuang* strain analyzed by Liu et al. [85] or *S. sanghuang KangNeng* studied by Shao et al. [86] would have been the first confirmed strain of mushroom whose fruiting bodies, beyond any doubt, are able to synthesize flavonoids. It is important to use the particular strain/isolate in both types of experiments due to differences in metabolites and pigment synthesis among the strains of particular species. Such differences connected to melanins, *Monascus*-like pigments or carotenoids were observed in both microscopic [87,88] and macroscopic fungi [89,90].

Among publications concerning flavonoid synthesis in mushrooms, another group of papers can be distinguished. In this group, flavonoid presence in fruiting bodies was proven by HPLC methods, but there was no information about the culture media. An example of such an article is the paper published by Alam et al. [91]. The authors conducted experiments on *Pleurotus nebrodensis* fruiting bodies by applying an HPLC-DAD system, and flavonoids such as hesperetin, naringenin, formononetin and biochanin-A were detected. Mushrooms were obtained from the Mushroom Research Institute of Gyeonggi Province in Korea, but there was no information about the cultivation media/spawn, and thus, the possible absorption of flavonoids or the stimulation of the production of the compounds caused by interactions with other organisms cannot be excluded. Thus, it is strongly suggested to include detailed information about the cultivation media composition and culture conditions in all studies dedicated to flavonoid synthesis in the fruiting bodies of mushrooms.

## 4. Additional Suggestions

Experiments on the flavonoid content in the cells of fruiting bodies of mushroom strains may be conducted using various schemes depending on the aim. If mushrooms are cultured for sale and consumption purposes, there is no need to verify whether the compounds are absorbed or synthesized, and thus, the HPLC analysis of fruiting bodies grown on any sort of media/spawn seems to be sufficient. However, if the ability to synthesize flavonoids is analyzed for scientific purposes, more parameters should be taken into consideration, e.g., culture media and cultivation conditions. In this case, the analysis of the flavonoid content in fruiting bodies grown from pure cultures cultivated on media/spawn free of flavonoids seems to be the best option. If fruiting bodies cannot be cultivated on media of controlled composition and specimens are obtained as environmental samples, a metabolomic analysis should be combined with the analysis of gene expression in particular developmental stages of fruiting bodies. It is important to remember that indicating only the presence of genes responsible for flavonoid synthesis is not sufficient to prove the ability to synthesize the compounds due to the possible lack of expression of these genes. In order to obtain strong and unequivocal evidence of whether mushroom fruiting body cells are capable of synthesizing flavonoids, we propose a scheme showing the selection of methods that should be used in experiments concerning environmental and laboratory-grown samples (Figure 2). Additionally, the results would not be confounded by the possible influence of neighboring organisms, such as plants, microorganisms and other mushrooms, on the stimulation of flavonoid biosynthesis. The proposed scheme of analyses could also be applicable for studies of compounds other than flavonoids in mushrooms.

## 5. Summary

The discussion of the capability of mushroom cells to produce flavonoids has been ongoing for years. In 2016, Gil-Ramírez et al. [33] published an important article showing the analysis of 24 mushroom species, and their conclusion was that the mentioned fungi are not able to synthesize or absorb flavonoids. In addition, searching in bioinformatic databases has shown no mushrooms with the gene sequences of all three enzymes postulated as essential for flavonoid biosynthesis, which are Pal, Chs and Chi. However, the statement that mushrooms do not contain flavonoids seems to be unnecessarily extended to all species and strains. In our opinion, the publication by Gil-Ramírez et al. [33] started the discussion of whether mushroom cells are capable of synthesizing flavonoids and the methods that should be used or avoided to verify this issue. The results of applying colorimetric analyses [33] led to the conclusion that more sophisticated methods, such as those using HPLC systems, should be involved instead of UV-Vis measurements only. Other papers, such as the article by Mohanta [34], which was a response to Gil-Ramírez et al.’s [33] paper, provided interesting information about mushrooms and fungi in general in the case of flavonoid synthesis, but they did not prove flavonoid synthesis inside the cells of mushroom fruiting bodies. Meng et al. [79] and Wang et al. [80] used both instrumental methods and gene expression analyses and showed that the cells of *A. cornea* and *S. baumii* mushrooms are able to synthesize flavonoids; however, mycelia, but not fruiting bodies, were studied. The results provided by Liu et al. [85] proved that the fruiting bodies of some strains of *S. sanghuang* collected as environmental samples contained flavonoids, but there was no information on whether the genes associated with flavonoid biosynthesis were expressed; thus, the detected compounds could be synthesized by the mushroom or absorbed from the host mulberry tree. On the other hand, studies conducted by Shao et al. [86] revealed that genes engaged in flavonoid biosynthesis were expressed in the developmental stages of fruiting bodies of *S. sanghuang KangNeng* cultivated in the lab, and the lack of confirmation of flavonoid presence in fruiting-body cells is the only reason why the synthesis of flavonoids in the analyzed mushroom strain was not fully proven. However, the results obtained by Liu et al. [85], Shao et al. [86] and Wang et al. [80] make the cells of the fruiting bodies of some strains of the *Sanghuangporus* genus highly probable flavonoid producers. Other works, such as Alam et al. [91], revealed flavonoid content in mushroom fruiting bodies but did not provide information about the culture media and culture conditions; thus, similarly to environmental samples, the absorption of the compounds by fungal cells cannot be excluded.

In conclusion, as can be found in the cited articles, although many researchers have been close to resolving the mystery of flavonoid synthesis in mushroom fruiting-body cells, to the best of our knowledge, the veil of the mystery over this issue has not been fully lifted yet.

Studying the problem is technically challenging; nevertheless, we propose factors that should be taken into consideration to conduct experiments to provide credible results. Firstly, one particular strain/isolate should be studied. Secondly, exclusively fruiting bodies, not mycelia, should be analyzed. Thirdly, if mushroom fruiting bodies can be cultivated on media with a well-defined composition, instrumental analyses, for example, HPLC analysis, would be sufficient to prove flavonoid synthesis. When fruiting bodies are environmental samples, a combined analysis using instrumental methods and gene expression studies should be applied. The suggestions made are clear signposts on the paths leading to the definitive verification of the ability of mushroom fruiting body cells to synthesize flavonoids, and the goal itself already seems very close to being reached.

## Figures and Tables

**Figure 1 cells-11-03052-f001:**
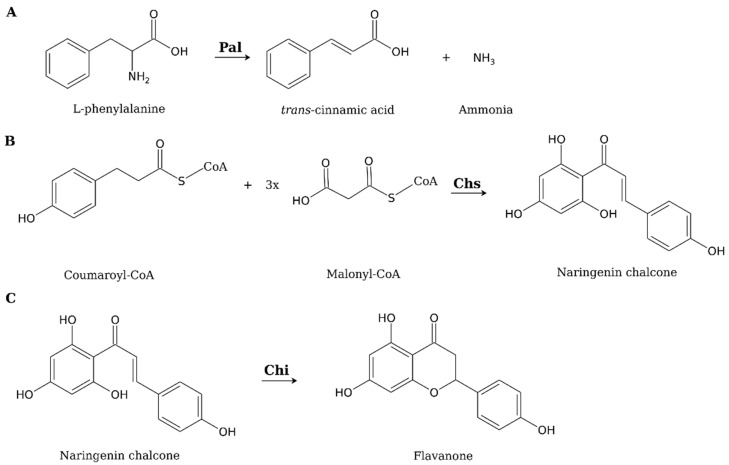
Reactions catalyzed by: phenylalanine ammonia-lyase (Pal)-(**A**), chalcone synthase (Chs)-(**B**) and chalcone isomerase (Chi)-(**C**). Based on [66] and modified.

**Figure 2 cells-11-03052-f002:**
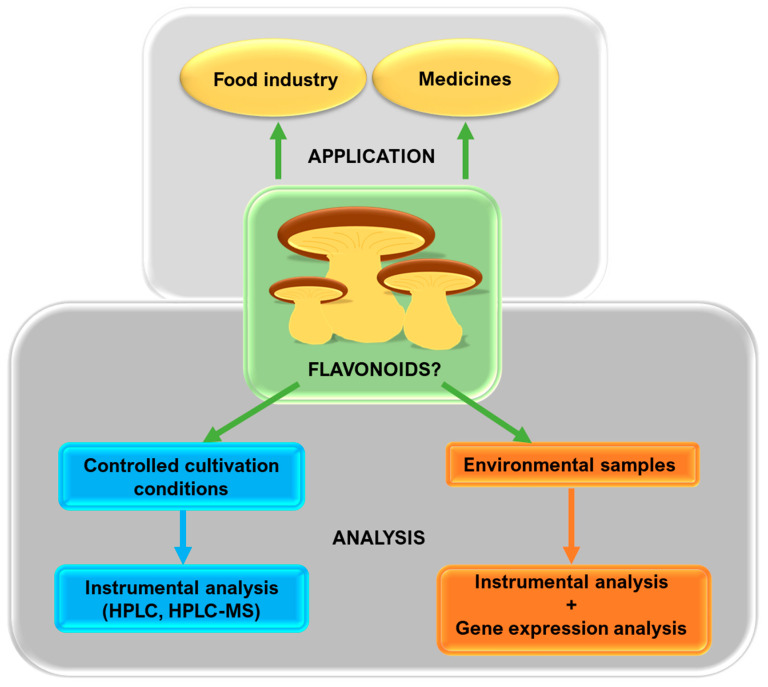
Proposed scheme of experiment selection for different specimens.

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
