# Peer review of "Secrets of Flavonoid Synthesis in Mushroom Cells"

_cells, 2022, doi:10.3390/cells11193052_

Round 1
Reviewer 1 Report
Generally, this paper is well organized and easy to follow. It’s suitable to publish before minor revision.
1) Line 8 i.e., usually not recommend to use short names in the abstract. And here it is not necessary to add i.e., and will confuse the readers.
2) In the introduction, the authors presented a large paragraph on fungi cells. In my opinion, maybe just focus on mushroom cells?
3) Are mushrooms capable to produce flavonoids? When I read this section, I am very looking forward to the author’s direct answer in the end. Since it is a interrogative sentence, maybe it’s better for the authors to response this question in the end.
4) I don’t know why some words should be in capital, such as line 187 The mentioned “Author”, line 226 The “Researchers”, line 235 in “Fungi Kingdom”
5) I feel that some sentences may be needed to improve. If possible, it’s better to have English natives to read it.
Author Response
Dear Reviewer 1,
We kindly thank you for your acceptance to be a reviewer and for your precious time spent on reading and analysing our manuscript. We are also grateful for your kind and positive evaluation of our work. We appreaciate your suggestions and have found them very valuable. According to them, we have made corrections which would definitely improve our manuscript. Below we present our answears to your suggestions point by point.
- ‘Line 8 i.e., usually not recommend to use short names in the abstract. And here it is not necessary to add i.e., and will confuse the readers.’
Response: Thank you for your suggestion. We have corrected the sentence as follows: ‘They are considered as valuable food additives of pro-health properties, and their sources have been searched also in other kingdoms’ (lines: 7 – 9).
- ‘In the introduction, the authors presented a large paragraph on fungi cells. In my opinion, maybe just focus on mushroom cells?’
Response: We kindly thank you for your suggestion. However, due to the ambiguity of the term 'mushroom' and the great confusion in the use of this term (e.g. some sources define mushrooms as only fruiting bodies and others as species producing fruiting bodies and mycelia), we have decided to present firstly to the reader the information on fungal cells using a term that is also specific to mushrooms, as they are made of fungal cells too. It would have been safer and closer to the truth to separate the fruiting body cells from the mycelial cells in the description, however, as our literature review has shown, we would then have had to write a completely different and very extensive review (maybe we will try it one day). Now, in order to avoid confusion resulting from different meanings of the same term, we have presented the properties of fungal cells, using a term unambiguously understood by all, in the introduction. Appreciating the validity of your remark (thank you again), we emphasised the fact that the term mushroom is understood differently by adding in the revised version of the manuscript the sentence ‘There are two definitions, the first one describes mushrooms as macroscopic fruiting bodies only, and the second, in our opinion the better one, as species producing macroscopic fruiting bodies and mycelia growing through nutrient substrate’ (lines: 55 – 58). We also drew attention to this in the abstract by modifying the sentence: ‘Mushrooms are usually defined as a group of fungi species capable to produce macroscopic fruiting bodies and there are many articles considering content of flavonoids in this group of fungi.’ (line 10).
- ‘Are mushrooms capable to produce flavonoids? When I read this section, I am very looking forward to the author’s direct answer in the end. Since it is a interrogative sentence, maybe it’s better for the authors to response this question in the end.’
Response: We are grateful for your statement! I think our feelings in searching for the answer to the question of whether mushrooms produce flavonoids were exactly as you have presented here. Digging through piles of articles, wanting to remain scientifically literate, it is impossible to conclude unequivocally whether mushrooms synthesise flavonoids. It was this conclusion that prompted us to write this paper, because it is all too unbelievable and odd that in the 21st century, with such advanced research technology, we cannot, as researchers, answer a relatively simple question unequivocally and confidently. We hope that this review will broaden awareness of this lack of knowledge and accelerate the knowledge of the answer. Thanking you for your attention, we have added the following sentence at the end of the section ‘Are mushrooms capable to produce flavonoids?’: ‘After reviewing the two mentioned articles the question whether mushrooms are capable to produce flavonoids remained still unanswered.’' (lines: 219 – 220) because you are absolutely right - if there is a question there must be an answer. Thank you!
We would also like to add, that the groundbreaking articles of Researchers [79,80] mentioned in the text later, proved beyond any doubt, that mycelial cells of mushrooms are capable to synthesise flavonoids ‘Despite the not fully understood biosynthetic pathways of mushroom flavonoids, Meng et al. [79] and Wang et al. [80] proved the capability of mycelial cells of mushrooms to produce flavonoids’ (lines: 341 – 343).
- ‘I don’t know why some words should be in capital, such as line 187 The mentioned “Author”, line 226 The “Researchers”, line 235 in “Fungi Kingdom”’
Response: We kindly thank you for your question. The capital letters in ‘Author’ or ‘Researchers’ were used out of respect only. We know it is not obligatory, however we did it to refer to the mention Scientists in more polite manner. In case of ‘Fungi Kingdom’ we have found in many sources, that capital letters were used as well as lowercase letters. However, according to your question and Brittanica encyclopedia we decided to use lowercase letter in the words ‘kingdom’ and capital letter in ‘Fungi’ or ‘Plants’. Once again, thank you for your constructive question!
- ‘I feel that some sentences may be needed to improve. If possible, it’s better to have English natives to read it.’
Response: Thank you for your suggestion. We have had the text proofread by a native speaker and modified according to her suggestions.
Yours sincerely,
Dariusz Latowski
Author Response
Dear Sir/Madam (Reviewer 2),
We kindly thank you for your acceptance to be a reviewer and for your precious time spent on reading and analysing our manuscript. We appreaciate kind and positive evaluation of our work. We are also grateful for your comments in the first part of the review. We would like to add, that we also admire your article and we consider your results as very significant and valuable. We appreaciate your suggestions and we have found them very constructive. According to them, we made corrections which would definitely improve our work. Below we present our answers to your suggestions point by point.
- ‘to my opinion it is more scientific to define a ́mushroom ́ (although it might not be necessary) using a taxonomy book more than a general encyclopedia.’
Response: Thank you for your suggestion! We have been looking for the definitions in many scientific sources and generally, there have been two different definitions: mushrooms as only fruiting bodies or as species producing fruiting bodies and mycelia. Finally we chose the second option as a better one, and as a source we pointed out Chang & Miles 1992 [26] article, which has been cited in, for example, book ‘Mushrooms as Functional Foods’ (Wiley, 2008). As term ‘mushroom’ is not related to taxonomic category and even in scientific community opinions are not consistent, we decided to present encyclopedical definitions, because it is highly probable that also many scientists use these sources. However, your suggestion inspired us to add a source from scientific literature, describing the first option (mushroom as fruiting body only). We hope that now, thanks to your suggestion, in the revised version of the manuscript, the issue raised is presented more clearly and scientific. Additional sentence is ‘Also some typically scientific sources define mushrooms as fruiting bodies only, for example ‘Mushrooms are macroscopic fruiting bodies produced by ascomycete and basidiomycete fungi during their sexual reproduction cycles’’ (lines: 68 – 70).
- ‘The references to carotenoids (as well as level of proteins etc.) are unnecessary since flavonoids and carotenoids followed totally different biosynthetic paths in fact, they are completely different molecules and not all the flavonoids are pigments.’
Response: We kindly thank you for your suggestions. We fully agree, that flavonoids are completely different molecules than carotenoids or proteins and biosynthetic pathways of the compounds are different. However, we wanted to emphasise, why mushroom fruiting bodies are attractive for consumers, why they are important and then, after proving their valuability, we pointed that they might also be a source of flavonoids known of pro-health properties. We also wanted to emphasise the scientific nature and importance of the term mushroom in this way. In our narration, we mentioned about pigments production to add additional capability of mushroom fruiting bodies and to prepare introduction for describing flavonoids. We fully agree that not every flavonoid is a pigment. Thanks to your comment we modified sentence into ‘As it was mentioned earlier, pigments content, such as carotenoids and part of flavonoids (since not every flavonoid is a pigment), as food ingredients, is desirable in fungi cells.’ (lines: 84 – 86). Thank you!
- ‘I did not understand some comments for instance on Monascus-like pigments, perhaps the text needs rewriting of the sentences.’
Response: Thank you! We wanted to emphasise, that among strains/isolates of one species, metabolite profiles may be different, as examples we pointed pigments such as melanins, Monascus-like pigments and carotenoids. In case of Monascus-like pigments, that term describes pigments similar to those synthesized by fungi from the genus Monascus (Monascus pigments – MPs). Monascus-like pigments may be produced by fungi from genera other than Monascus. The term was used in cited article [87] in our manuscript, and was used also in https://doi.org/10.3389/fmicb.2018.03143. However, thank to your question, we realised that Monascus should be written in italics. We hope that this amendment will make this clear to the reader. Thank you!
- ‘To my personal opinion passive voice in some verbs and impersonal sentences make the document less subjective. Figure 2 is unnecessary, and the title is for me too pretentious or too poetic it could be more ́scientific ́.’.
Response: We are grateful for this suggestion. In our opinion, Figure 2 makes our statement clearer. The need for both genetic and instrumental analyses for environmental samples is one of our most important practical considerations in this paragraph or even in the whole manuscript. The great confusion with the assessment of flavonoid production capacity by mushrooms is due, as we try to show in the paper, to the lack of a holistic view of all possible technical (methodological) options. We leave Figure 2 as a pictorial summary of what can be done and how it can be done today to solve the puzzle of flavonoid synthesis by mushrooms.
As for the title, our intention was to be intriguing, to go a bit beyond strictly scientific language, among other things, because the crux of the problem addressed by the review also stems from a certain imprecision of colloquial terms with scientific ones, but also ambiguity in scientific language. We keep in mind here the term ‘mushroom’. Please accept our title proposal. In our opinion, it arises curiosity, but most importantly, it captures all the aspects of the work covered, everything related to the synthesis of flavonoids in mushrooms, which we find very difficult to define concisely with another title.According to your suggestion, we changed passive voice and transformed sentences in some fragments, for example in ‘Additionally, we suggest to analyse different developmental stages of fruiting bodies to indicate particular stage, in which flavonoid synthesis and accumulation occurs or is the most efficient.’ (lines: 314 – 316), ‘During literature review preparing two articles concerning flavonoids presence in Inonotus sanghuang (recently renamed as Sanghuangporus sanghuang), asian medicinal mushroom, especially drew our attention’ (lines: 317 – 319), ‘In order to obtain strong and unequivocal evidence whether mushroom fruiting body cells are capable to synthesize flavonoids, we proposed a scheme showing selection of methods, which should be used in experiments concerning environmental and grown in the lab samples’ (lines: 383 – 386). Thank you once again!
- ‘Author should also make a comment or take into consideration that flavonoids are compounds from the secondary metabolism of plants but mushrooms also have their own secondary metabolism and no paragraph indicating how they might build up flavonoid molecules is proposed, only a vague reference to fungal melanins is indicated but not how it might derive to flavonoid production by that ̈fungal biosynthetic pathway different than plants ́.’
Response: Thank you for your suggestion. We added fragment ‘In the second article by Shao et al. [86] expression of genes associated with secondary metabolism of Sanghuangporus sanghuang KangNeng strain. Authors studied four different developmental stages of the mushroom: 10 and 20-days old mycelia as well as 1 and 3-years old fruiting bodies. The Researchers analysed genes involved in biosynthesis of polysaccharides, triterpenoids and flavonoids’ (lines: 321 – 326) which should complement the text and emphasise, that fungal secondary metabolism may be a source of many interesting biocompounds. We also added fragment according to present knowledge of enzymes in biosynthetic pathways of flavonoids in mushrooms ‘Due to a variety of flavonoids detected in mushrooms mycelia [79,80], where the risk of absorption of the compounds could not appear, it is especially interesting what are the biosynthesis pathways of the compounds. Meng et al. in the genome of A. cornea found four homologous genes of pal, three homologues of polyphenol oxidase encoding gene (ppo) and two genes homologous to chi [79]. Wang et al. [80] during studies of S. baumii detected eighty-one flavonoids, but only four genetical sequences homologous to sequences encoding plants’ enzymes involved in flavonoids biosynthesis: pal, chi, 4-coumarate-CoA ligase (4cl) and isoflavone reductase (ifr) were found. The lack of for example sequences of cinnamate 4-hydroxylase (c4h), flavanone-3-hydroxylase (f3h) or flavonol synthase (fls), as well as chs, led the Researchers to the conclusion that in the case of mushrooms, in flavonoids biosynthesis other enzymes of the same superfamilies as the lacking ones are involved. The Authors also revealed that overexpression of sbpal increased the amount of flavonoids accumulated in mycelia [80].’ (lines: 282 – 294) and ‘The Researchers analysed genes related to flavonoid biosynthesis, and found sequences of chalcone isomerase 1 (chi1), flavonoid-3’, 5-hydroxylase (f3′5′h), f3′H -flavonoid-3′-hydroxylase (f3′h), flavanone-3-hydroxylase 1 (f3h1), flavanone-3-hydroxylase 2 (f3h2) and flavonol synthase (fls). What is worth to mention here, f3h homologues and fls were not found in the genome of S. baumii [80].’(lines: 327 – 332).
- ‘The two initial papers are summarized in detail and the new coming papers with interesting outcomes are only briefly described.’
Response: We are grateful for this comment! The reason why the first two publications were discussed in details was the fact, that they were controversial and became the reason for more in-depth study of the issue raised by other researchers, and inspired us to write this review. In our opinion both works contributed something positive and negative, thus we wanted to analyse all aspects of those articles step by step to point positive and negative aspects. Other publications were also very important, moreover, we found the experiments as well planned and results as credible. That is the reason why they were described briefly with the most outstanding and essential points. In these papers statements are well defended and there is no need to analyse whether they are valuable, without any doubt they are. In our opinion four articles ‘Transcriptomic and non-targeted metabolomic analyses reveal the flavonoid biosynthesis pathway in Auricularia cornea.’ (Meng et al., 2022), ‘Mushrooms do produce flavonoids: metabolite profiling and transcriptome analysis of flavonoid synthesis in the medicinal mushroom Sanghuangporus baumii’ (Wang et al., 2022), ‘Polyphenolic composition and antioxidant, antiproliferative, and antimicrobial activities of mushroom Inonotus sanghuang.’ (Liu et al., 2017) and ‘The Genome of the Medicinal Macrofungus Sanghuang Provides Insights Into the Synthesis of Diverse Secondary Metabolites.’ (Shao et al., 2020) provided the most interesting information in the field of flavonoids production by mushrooms. However, your comment encouraged us to add some additional information from mentioned articles, such as: ‘They also revealed, that addition of leachate of Korshinsk peashrub (Caragana korshinskii) to the complete yeast medium (CYM) could increase the flavonoids amount up to 23.6% in comparison with amount obtained after cultivation of A. cornea on control CYM medium [79].’ (lines: 270 – 273), ‘The Researchers applied UPLC–MS/MS analysis and identified eighty-one flavonoid compounds from nine groups: flavanols, flavanonols, flavones, flavonols, flavanones, chalcones, isoflavones, isoflavanones and proanthocyanidins in numbers of 10, 4, 25, 23, 5, 2, 8, 1, 3 respectively. Flavanonol pinobanksin, glycosylated and acylated forms of isoflavone genistein were detected and flavonol kaempferol linked to pro health properties of S. baumii, as well. Wang et al. [80] also revealed, that increased aeration time promoted synthesis of flavonoids, what would be especially valuable in case of industrial production of the compounds.’ (lines: 275 – 282), ‘In the second article by Shao et al. [86] expression of genes associated with secondary me-tabolism of Sanghuangporus sanghuang KangNeng strain. Authors studied four different developmental stages of the mushroom: 10 and 20-days old mycelia as well as 1 and 3-years old fruiting bodies. The Researchers analysed genes involved in biosynthesis of polysaccharides, triterpenoids and flavonoids. Interestingly, they suggested that mycelial stages, due to gene expression analysis, are more effective flavonoids producers than fruiting bodies.’ (lines: 321 – 327).
Thank you again for your work in improving our manuscript!
Yours sincerely,
Dariusz Latowski